

# How to use mixed precision in Ocean Models

Oriol Tintó Prims[1, 2], Mario C. Acosta[1], Andrew M. Moore[3], Miguel Castrillo[1], Kim Serradell[1],
Ana Cortés[2], and Francisco J. Doblas-Reyes[1,4]

[1]Earth Sciences Department, Barcelona Supercomputing Center - Centro Nacional de Supercomputación, Barcelona, Spain
[2]HPCA4SE research group, Computer Architecture and Operating Systems Department, Universitat Autònoma de Barcelona,
Bellaterra, Spain
[3]Ocean Sciences Department, University of California, Santa Cruz, CA, USA
[4]ICREA, Barcelona, Spain

**Correspondence:** Oriol Tintó (oriol.tinto@bsc.es)

**Abstract.** Mixed-precision approaches can provide substantial speed-ups for both computing- and memory-bound codes requiring little effort. Most scientific codes have overengineered the numerical precision leading to a situation where models are using more resources than required without knowing where these resources are unnecessary and where they are really needed. Consequently, there is the possibility to obtain performance benefits from using a more appropriate choice of precision and the

only thing that is needed is a method to determine which real variables can be represented with fewer bits without affecting the accuracy of the results. This paper presents a novel method to enable modern and legacy codes to benefit from a reduction of precision without sacrificing accuracy. It consists of a simple idea: if we can measure how reducing the precision of a group of variables affects the outputs, we can evaluate the level of precision this group of variables need. Modifying and recompiling the code for each case that has to be evaluated would require an amount of effort that makes this task prohibitive. Instead,

the method presented in this paper relies on the use of a tool called Reduced Precision Emulator (RPE) that can significantly streamline the process . Using the RPE and a list of parameters containing the precisions that will be used for each real variable in the code, it is possible within a single binary to emulate the effect on the outputs of a specific choice of precision. Once we have the potential of emulating the effects of reduced precision, we can proceed with the design of the tests required to obtain knowledge about all the variables in the model. The number of possible combinations is prohibitively large and impossible

to explore. The alternative of performing a screening of the variables individually can give certain insight about the precision needed by the variables, but on the other hand some more complex interactions that involve several variables may remain hidden. Instead, we use a divide-and-conquer algorithm that identifies the parts that cannot handle reduced precision and builds a set of variables that can. The method has been put to proof using two state-of-the-art ocean models, NEMO and ROMS, with very promising results. Obtaining this information is crucial to build afterwards an actual mixed precision version of the code

that will bring the promised performance benefits.

## 1 Introduction

Global warming and climate change are a great challenge for human kind, and given the social (i.e. see Kniveton et al., 2012 about climate refugees), economic (i.e. see Whiteman G, Hope C, 2013 about the trillion dollar problem) and environmental



threat (i.e. see Bellard et al., 2012 about mass extinctions) that it poses, any effort to understand and fight against it falls short. A better knowledge and greater capacity to forecast how the climate will evolve can be a game changing achievement, since it could help to justify ambitious policies and help to adapt for future scenarios (see Oreskes et al., 2010). The Earth System is an amalgamation of many parts: the atmosphere, the hydrosphere, the cryosphere, the land surface and the biosphere. All

these elements are extremely rich in phenomena, open and inter-related, interchanging fluxes of mass, heat and momentum in ways that are virtually endless, some of which are poorly known or perhaps even unknown. The magnitude and complexity of these systems make it difficult for scientists to observe and understand them. For this reason, the birth of computational science was a turning point, driving to the development of Earth System Models (ESMs) that allowed experiments that otherwise were impossible. ESMs, despite being incomplete, inaccurate and uncertain, have been a framework where it is possible to build upon

knowledge, and have become crucial tools (see Hegerl and Zwiers, 2011). Since inception, the capacity to mimic the climate system has increased, and with it the capacity to perform useful forecasts (see Bauer et al., 2015). The main developments that have led to this improvements in model skill are the improvement of the physical parameterizations, the use of ensembles, the improvement of the model initialization and increases in resolution (see Bauer et al., 2015), most of them contributing to an increased computational cost (see Randall et al., 2007). For this reason, these developments are only possible by an increasing

availability of computing power, a situation that will continue into the future. The motivation to make models efficient is twofold. On one hand, developments that are considered crucial to advance the skill of the models require more computational power. This is not just a matter of having a larger machine since some of the issues that emerge are not trivial and require additional developments (see Dennis and Loft, 2011). On the other hand, the huge investment in computational resources that is necessary to perform simulations with ESMs models implies that investing time in optimizing them will pay-off.

One research field that gained momentum and that can improve model performance is the use of mixed precision algorithms. Until not so long ago, the speed of most computations was constrained by how fast a CPU could perform operations, with the speed of the memory being fast enough to provide more data than the processor could process. In addition CPUs were designed in a way that they could virtually perform operations at the same speed no matter whether they were operating with 32-bit or 64-bit floating point representations. Therefore, the only benefit of using less precision to reduce the memory requirements and

the computational performance was not so much a motivation. Mainly two factors changed that scenario. First, the CPU speed increased at a faster rate than memory speed implying that at some point many codes that before were CPU-bound would become memory-bound, with the memory-bandwidth being insufficient to feed all the data that the processor can process. Second, vector operations doubled the number of floating point operations per cycle that could be performed when the number of bits of the representation is halved. For this reason, now the use of smaller representations provide performance benefits

that justify the effort of optimizing the numerical precision (see Baboulin et al., 2009), and while this is true with the actual hardware, the expected potential is even bigger in future architectures that will include not only 64-bit and 32-bit but also 16-bit arithmetic.

*We are now in a situation where ESMs, as computer codes of other domains, need to use computational resources efficiently and where mixed-precision approaches emerge as a potential solution to help improve efficiency.*





The main risk of reducing the precision is falling short, since using less precision than needed can led to numerical errors making model results inaccurate or directly wrong. The precision that is required depends on many factors, so what is needed is a method to identify which variables can effectively use less precision without compromising the quality of the simulations and which ones do require high precision. If there is room to exploit the reduction of precision it is generally because in

many situations the precision has been over engineered. One example is the precision used to represent the input data fed to ESMs. The precision of the physical observations is limited by the instruments used to make them. In the case of sea surface temperature measurements from earth orbiting satellites, this precision is about a few tenths of degree. To represent this information 16-bit floating point representation could be enough but, instead, 64-bit representations are often used.

Recent work has demonstrated the potential benefits that mixed-precision approaches can provide to many different kinds of
codes, since it is possible to achieve substantial speed-ups for both computing- and memory-bound codes requiring little code effort (see Baboulin et al., 2009). The spectrum of studies go from explicit code manipulation in very specific algorithms to automatic modification of binaries of any kind of code. Some studies have focused on the use of mixed-precision algorithms to obtain performance benefits without compromising the accuracy of the results (see Baboulin et al., 2009). There are also several automatic mixed-precision exploration tools (see Graillat et al., 2016 Lam et al., 2012) that have been mainly tested on small

benchmarks, usually C++ codes. These kinds of studies inspired Earth Science groups working with ESMs that are willing to improve their computational performance to make bigger and more ambitious experiments possible (see Váňa et al., 2017, Düben et al., 2014, Düben et al., 2017 and Thornes, 2016). Inspired by previous works, we propose a method to automatically explore the precision needed for the real variables used in state-of-the-art ESMs.

The method emerged when trying to explore how to achieve simulations as similar as possible to the standard double
precision simulations while reducing the precision of some of the variables used. Our work extends the research mentioned above to achieve mixed precision implementations for full-scale models. To do so we rely on the Reduced Precision Emulator (RPE) (see Dawson and Düben, 2017), that mimics the effects of using an arbitrary number of significand bits to represent the real variables in a code and measure the impact of a specific reduced precision configuration in the output produced by the model. Minimizing the user intervention by automating all the tedious intermediate processes makes possible an analysis of

models that otherwise would require too many human resources.

To prove the methodology, this work includes two cases studies. These cases correspond to two different ocean models that are widely used worldwide: NEMO and ROMS. With these models we demonstrate how we can use the methodology with different applications, thus demonstrating its potential.

## 2   Method

In this section we will show how we can establish which real variables in a Fortran code can effectively use less precision than the *de facto* standard 64-bits. The reader will find an explanation about why and how we developed this method, with the specific steps of the methodology detailed below. The basic idea behind the method is to perform simulations with a Fortran model using a custom set of precisions and directly assess the outputs to see if the results are accurate enough. To do so we



use the RPE tool, that is a Fortran library that allows us to simulate the result of a floating point operation given that the variables are using a specific number of significand bits. This can be integrated into an actual code to mimic what would be the consequences of using reduced precision in certain variables. The strength of the tool is the flexibility that it offers once the tool has been implemented into the code, allowing us to easily test any given combination of precisions. The main drawback is

the considerable overhead added to the simulations, increasing its cost.

The objective of the method is to find a set of precisions that minimizes the numerical resources used keeping the accuracy of the results, meaning that they are arbitrarily close to the results obtained using high numerical precision. A set of precisions is a specific combination of the precisions assigned to each variable, being the number of possible sets $52^n$ , where $n$ is the number of real variables used in the code and 52 the number of bits used to describe the significand in a double-precision representation.

This number makes it prohibitively expensive to explore all the combinations. For any real-world code, this is true even not considering all the possible values between 0 and 52 but just considering double-precision (64-bits), single-precision (32-bits) and half-precision (16-bits), where the number of possibilities is still $3^n$.

A feasible alternative to testing all possible combinations is to perform a screening of the variables individually. The idea is to simply perform $n$ simulations to observe what happens when all the variables are retained at high precision except the one

that is being tested. While it is true that this approach can give an insight about the precision needed by the variable, it has some drawbacks. Using this approach most of the variables that do require high precision can be identified, but many issues regarding more complex interactions between variables may remain hidden. Besides, building a complete set of variables from the tests performed on individual variables is not trivial at all i.e. a set formed by variables whose individual test concluded that could use lower precision safely can end up producing non-accurate results.

Another alternative, the method which we propose in this article, is a divide-and-conquer algorithm that identifies the parts of the code that cannot handle reduced precision and builds a complete set of variables that can. Starting from a variable set that contains all the real variables that we want to analyze, the approach consists of evaluating what happens when the set uses reduced precision. If the results become inaccurate, we proceed to split the set in two parts, that are evaluated individually. The process is recursive and ideally a binary search is performed until the sets that are evaluated contain only a single variable. If

a set containing a single variable ends up being inaccurate, this variable is kept in high precision in the preceding sets, that are re-evaluated excluding those variables. The advantage of this approach is that it is cheaper to find all the variables that can individually compromise the results, and in addition it is easier to rebuild a complete set assimilating the results of the simulations of subsets that gave an accurate result.

Nevertheless, there are some elements that prevent this approach from working properly. The non-linearity of most of the

Earth science codes inevitably implies that the differences that appear between two simulations performed using different numerical precisions will not be constant. A single measure of this difference does not contain any information about its variability, so we can only tell whether a specific case was accurate but not how likely it is that any other case will be accurate or not. One of the consequences is that in many cases where two subsets were accurate resulted in an inaccurate set when combined. To increase the confidence in the results even considering the stochastic behaviour of the numerical error, we

propose to re-evaluate the sets whose results are accurate with different initial conditions to account for its variability. The



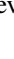

argument behind this procedure is that if an inaccurate result was obtained, we already know for sure that this specific set using reduced precision can in some instances produce inaccurate results, but if otherwise the results were accurate, we can not for sure consider that they will always be. This method is similar to the ensemble simulation used in ESMs (see Palmer et al. 2005), which tries to assess the uncertainty of the simulation outcomes by taking into consideration the uncertainty in the

model inputs. The method remains the same adding an extra simulation initialized with different initial conditions for the sets that show accurate results. In practice, the number of tests could be set higher but according to our results, an ensemble of two members already solved most of the issues related with combinations of accurate subsets resulting in inaccurate sets.

The steps of the methodology, which are discussed in the next subsections, consist of:

– Implement the emulator into the code completing all the necessary actions to obtain a code that uses the emulator in
which it is possible to select the precision of each real variable through a list of parameters.

– Establish a test that will determine if the results of a simulation are accurate enough.

– Perform a precision analysis by launching the necessary tests to obtain a set of variables that can effectively use reduced precision passing the accuracy test.

### 2.1 Implementing the emulator

The **RPE** is a Fortran library that allows the user to emulate floating point arithmetic using a specific number of significand bits. It has the capability to emulate the use of arbitrary reduced floating-point precision. Its development was motivated by the need to explore mixed-precision approaches in large numerical models that demand more computational power, having in mind weather and climate fields, making the tool highly suitable for our purpose. The emulator is open source, can be accessed through git-hub, has documentation available and a reference paper by Dawson and Düben (2017) with more detailed
information.

Although the use of the emulator facilitates the testing of different precision configurations without recompiling, in large codes like ESMs the implementation of the emulator can carry more work than expected. The length, the large development time, the quantity of different developers and the lack of a rigid style guide can end up with a big number of exceptions that make it harder to fully automate the emulator implementation, requiring the user to solve the emerging issues.

Our implementation of the emulator has two different parts:

1. Replacing the declaration of the real variables with the custom type **rpe_var**.

2. Introducing a way to select the precision of the variables without requiring recompilation of the code.

### 2.1.1 Replace variable declarations

To use the emulator, the user has to replace the declarations of the real variables with the custom rpe_var type defined in the
emulator library, as it is explained with detail in the reference manuscript (see Dawson and Düben, 2017). Even though the idea is quite simple and can be straightforward for simple codes, the practical process in a complex and large state-of-the-art





ESM can present several minor issues that can add up to a considerable amount of work. This is a list of some of the specific issues that can be found implementing the emulator:

– All the real variables that were initialized at declaration time need to be initialized providing a derived type variable instead of a real, which requires modification of all these declarations.

– When a hard-coded real is used as a routine argument where the routine is expecting an rpe_var variable, it is necessary to cast this hard-coded variable into a rpe_var type variable.
(i.e. "call routine$(var, \mathbf{1.0}, var2, \mathbf{0.1})$" has to be "call routine$(var, \text{rpe\_var}(\mathbf{1.0}), var2, \text{rpe\_var}(\mathbf{0.1}))$" )

– Although it is possible to adapt the RPE library to include intrinsic functions (i.e. max,min) that can use rpe_var variables as arguments, there is still a problem when mixing rpe variables and real variables. The problem can be overcome
converting all the variables to the same type.

– When there is a call to an external library (i.e. netcdf, mpi), the arguments cannot be rpe_var and must be reals.

– Read and write statements that expect a real variable cannot deal with rpe_var variables.

– Several other minor issues (pointer assignations, type conversions, used modules,...)

In codes with hundreds/thousands of variables this task can represent months/years of work, and for this reason it is worth-
while to automate the whole process. When all the issues are solved the model should be able to compile and run.

### 2.1.2 Selecting the precision

To specify at runtime the precision of each individual variable, the method that we use is to create a new Fortran module which includes an array of integers containing the precision value of each one of the real variables of the model, that has to be accessible from every other module of the model. The values of this array will be set at the beginning of the simulation
by reading a list of parameters and when this is done, the final step is to assign these values to the actual variables. This can be done before it is first used or at specific points (i.e. after a routine call). For allocatable variables the assignment can be done trivially just after the allocation but there are more complicated cases where the variable is declared in a module but used by several other modules. A different case are the variables that are used as routine arguments (i.e. "call routine(variable)"), if inside the routine the variable has its precision modified, after the routine call it will be necessary to reestablish the proper
precision and truncate the result if needed. Again, this process can be tedious and only automating it can be useful for large codes.

After implementing the emulator into a code, one should be able to launch simulations specifying individually the number of significand bits used for each variable and obtain outputs, having completed the most arduous part of the proposed methodology.





## 2.2 Designing the accuracy tests

Once the emulator has been implemented and we are able to launch simulations, we must define how we will evaluate the accuracy of the results, the kind of experiment that we want to perform, and define a True/False test to perform on the outputs.

To define a test to discriminate between accurate or inaccurate results, we must define a function that can be applied to
simulation output and determine if the outputs are arbitrarily close to our reference in double precision or not.

To put a simple example, given a simulation whose only output is a single scalar value, we can consider that a given test is accurate if an arbitrary amount of significant digits of the answer are correct. Having the value of $\pi$ as a reference ($\pi = 3.14159265358979323846264338327950288...$), we can consider that a given simulation is accurate enough if the difference between the reference and the value obtained is smaller than a given threshold, i.e. the results coincide up to a specific number of
significant digits. Given the following result, $result = \mathbf{3.14159}3124125123124162123124521353 1$ , imposing as a condition to coincide in at least 10 significant digits, the outcome of the test would be **False**, if instead the condition is to coincide in at least the first 4 significant digits, the outcome would be **True**.

## 2.3 Performing the analysis

Following this methodology we propose a recursive divide and conquer algorithm with few slight modifications. The basic
approach that has been used as a starting point is the following: For a given set of variables, we generate a list of parameters that identify these variables to use the desired reduced precision and using these parameters we launch a simulation. When the simulation is completed, we proceed to apply the accuracy test described in section 2.2. If the simulation with the specific set passes the test we consider that it is safe to reduce the precision of the variables in the set, otherwise we consider that it is not safe. In this case, the purpose of the algorithm is to identify which part of the parameter set is responsible for the inaccuracy,
in which case we proceed by sub-dividing the set and evaluating its parts separately. In the case where the set of variables contains only a single variable, then we have identified one of the variables that needs to preserve higher precision.

The sets that yield inaccurate results and can be divided require information from the subsets after these have been evaluated to be modified and reevaluated. In case a subset fails, the variables from this subset will be discarded from the original set, while in cases that are deemed accurate, all the variables of the subset will be retained in the set, with the exception of those
variables that were discarded previously.

Notice that the procedure is recursive because any given subset will follow the same approach, until the point where the set contains a single variable and the source of the potential error can be isolated.

This initial approach has some drawbacks: Numerical error in most algorithms has a stochastic component which, combined with the non-linearity of this kind of models, can mean that a specific set of variables that under certain conditions gives
accurate results, under different conditions the results are not accurate anymore. The effect of reducing the precision of a set of variables can not be directly deduced from what happens to its parts. The simplest case where this can happen is when we deal with two variables that appear in the same arithmetic operation: using the same logic as the actual processors, the emulator performs intermediate operations using the largest precision between the variables involved, so it may happen that having any





of the two variables in higher precision will give an accurate enough result but not when both of the variables use reduced precision.

To illustrate this lets consider the following example: having two variables $x$ and $y$, with $x = 2^6$ and $y = 2^{-6}$, if we compute the sum $z = x + y$ in real arithmetic, the result is $z = 64.015625$. Both $x$ and $y$ can be perfectly represented using a 10-bit

significand, but it is not the case for $z$ that requires more numerical precision. Following the processor logic, if either $x$ or $y$ use a 52-bit significand, the computation of $z$ will be done using 52 bits leading to a correct result, but if otherwise both variables are using 10 bits, the computation will be done using 10 bits and will yield a wrong result ($z = 2^6$).

If the two effects individually already represent a challenge, both combined are an issue that may be difficult to solve. Looking deeper in to cases where two accurate sets when combined give inaccurate results, we see that reevaluating the

accurate subsets with different initial conditions, the variability was large enough to make the results inaccurate according to the defined test. For this reason we added an extra step into our algorithm that consists in an extra assessment of the results in case that the first evaluation resulted in a positive outcome. While we cannot be sure that performing only a single re-evaluation will be enough to ensure that the full spread of possible outcomes will fall under the desired threshold, it can be sufficient for the most sensitive cases.

To perform extra tests, we could have different initial conditions to perform simulations that are different and can give us information about the uncertainty of our results. The number of different initial conditions that have to be tested to consider the results as successful can be changed, keeping in mind that increasing the number of initial conditions will increase the total number of simulations that will be needed to finish the full workflow and identify an appropriate set.

Another slight modification that we added to the initial approach is the definition of stricter thresholds for smaller subsets,

to prevent the possibility that the errors add-up in bigger sets.

The whole algorithm can be described with the pseudo-code presented in Appendix 1, that contains the instructions that any given set has to follow in order to learn which elements of the set can use reduced precision.

## 3  Study cases

In this section we will present a proof of concept of the method using two state-of-the-art models, NEMO and ROMS. For

each one of the two models we carried out two different experiments. Using NEMO, we performed two analysis using different accuracy tests having as a target reduced precision single-precision (i.e. the variables in reduced precision are using a 23-bit significand). Using ROMS instead, we performed two analysis with the same accuracy tests but having two different reduced precision targets, single and half precision (i.e. in the firts of the analysis the variables in reduced precision are using a 23-bit significand while in the second one are using a 10-bit significand). The section is divided in three parts, having one part for

each model and a discussion of the results.



## 3.1 NEMO

NEMO (from Nucleus for European Modelling of the Ocean) is a state-of-the-art modelling framework of ocean related en-
gines. The physical core engines for the different parts of the model are the OPA engine for the ocean, LIM for the sea-ice and
TOP-PISCES for the biogeochemistry (see Madec 2008, Rousset et al. 2015 and Aumont et al. 2015). The range of applications

includes oceanographic research, operational oceanography, seasonal forecast and (paleo)climate studies. The model is used
by a large community of users since 2008, not less than 184 projects have been carried out and about 300 publications have
been published (NEMO development team 2018).

Previous performance analyses of NEMO have shown a flat profile where the most time-consuming routine was not even
responsible of a 20% of the computation time. For this reason, any effort to improve the computational performance of the

model needs to be something that does not target a single region of the code but something that can be applied along all the
sections. Following this approach some developments targeted the communications (see Tintó Prims et al., 2018), that are used
in almost all the parts of the model.

As explained in section 1, previous publications have demonstrated the positive impact that the use of mixed precision
approaches can have on the performance of scientific models. Previous experiences in reducing the working precision of

NEMO from 64-bit to 32-bit demonstrated a significant change in the results, indicating that blindly reducing the precision in
the whole code was not a valid possibility.

To make the outcome of this work relevant to the modeling community, the analysis has been performed using the version
4.0b of the code that at the time of writing is the latest version available. The configuration used was an ocean-only simulation
using the ORCA2 grid (about 220 km resolution near the equator) and the objective of the analysis is to identify which set of

variables can effectively use 32-bit floating point representations instead of the 64-bit standard while keeping the difference
with the reference below a chosen threshold.

### 3.1.1 Emulator implementation

We developed a tool that not only modifies the source code to implement the emulator solving all the issues mentioned in the
methods section, but also creates a database with information about the sources, including its modules, routines, functions,

variables and their relations. This database will have several uses afterwards, since it can be used to generate the list of
precisions assigned to each variable and to process the results of the analysis afterwards.

There is a specific issue regarding the use of routines or functions that are defined outside the code. In the case of NEMO,
these libraries are mainly MPI and netCDF. The MPI library is used to exchange information between processes and the netCDF
library is used for input/output of files in format netCDF, the used by the model. Generally, external routines can be handled

by passing them real arguments, however, there can be exceptions like some MPI routines that give additional problems. Since
there's no special interest in analyzing these routines, a simple solution is to keep these routines unmodified (without replacing
the real variables by derived types), and to do so we included in the tool the possibility to keep source files unmodified simply





by putting them in a list. The selection of the source files that should not be modified requires user expertise. After that the tool handles all the necessary workarounds to ensure that the proper variable type is passed as routine arguments.

### 3.1.2 Designing the accuracy tests

In this section we address how we can distinguish between accurate or inaccurate results, something that we need to be able to
do in order to proceed with the method. Our approach was to define a metric to evaluate how similar two simulations are and define a threshold above which we will consider simulations inaccurate.

The metric used to evaluate the similarity between two simulation outputs is the root mean square deviation divided by the Interquartile Range (RMSDIQR). The outputs of our simulation are 3D (2D + time) and 4D (3D + time) but treat the time dimension separately and compute the RMSDIQR of the 2D and 3D component respectively for each time-step and keep the
value of the maximum.

$$RMSD_{2D}(t)) = \sqrt{\frac{\sum_{i=1}^{i_{max}} \sum_{j=1}^{j_{max}} \left(ref_{i,j}(t) - x_{i,j}(t)\right)^2}{i_{max} \cdot j_{max}}} \tag{1}$$

$$RMSD_{3D}(t)) = \sqrt{\frac{\sum_{i=1}^{i_{max}} \sum_{j=1}^{j_{max}} \sum_{k=1}^{k_{max}} \left(ref_{i,j,k}(t) - x_{i,j,k}(t)\right)^2}{i_{max} \cdot j_{max} \cdot k_{max}}} \tag{2}$$

where $i, j, k$ are the spatial axis indices, $t$ is the time index, $ref_{i,j,k}(t)$ is the value of the reference simulation at a given point $i, j, k$ and a given time $t$ and $x_{i,j,k}(t)$ the value of simulation that is being evaluated at the same point $i, j, k$ and a given
time $t$.

$$IQR(t) = Q3(t) - Q1(t) \tag{3}$$

where $Q3$ and $Q1$ are the values of the third quartile and first quartile respectively, and $t$ is the time index.

So the final metric will be the accuracy score:

$$AccuracyScore = \max_{\forall t} \left(\frac{RMSD(t)}{IQR(t)}\right) \tag{4}$$

where:

$$RMSD(t) = \begin{cases} RMSD_{2D}(t) & \text{if case 2D} \\ RMSD_{3D}(t) & \text{if case 3D} \end{cases} \tag{5}$$

where the 2D cases are the variables that only cover a 2D domain like the sea surface temperature, and the 3D variables are those that cover the full 3D domain like the temperature.



The final accuracy test can be thus defined as:

$$AccuracyTest(AccuracyScore, threshold) = \begin{cases} \text{True} & \text{if } AccuracyScore < threshold \\ \text{False} & \text{if } AccuracyScore \geq threshold \end{cases} \tag{6}$$

To be able to use the test defined above with the outputs of our experiments, the elements required are a reference simulation and the thresholds. To obtain the reference simulation, first we have to define what we do want to compare to and the kind of test. For this analysis, we are trying to assess the accuracy of the outputs compared to the results that are obtained using the original version of the code with 64-bit real variables overall. The individual simulations consisted of 10-day simulations that produced daily outputs of the 3D temperature field, the 3D salinity field, the column integrated heat and salt content. Temperature and salinity were selected because these are the two active tracers that appear in the model equations. Reference simulations were launched for each different initial condition used later in the analysis.

In order to define some meaningful thresholds, we performed a simulation using a smaller time-step to compute the accuracy score against the reference simulation to use this value as a first guess to define our thresholds. For the four variables listed before, the F values for the different variables were $1.49 \times 10^{-3}$ for the temperature field, $6.41 \times 10^{-3}$ for the salinity field, $2.74 \times 10^{-3}$ for the ocean heat content and $5.47 \times 10^{-6}$ for the ocean salt content. To assess how defining different thresholds can led to different results and to emphasize that using this method it is possible to keep the magnitude of the perturbations arbitrarily small given certain conditions that need to be fulfilled, two different thresholds were defined (see Table 1). In the first of the two set of thresholds, we will define them to be one thousand times smaller than the differences obtained using a reduced time-step. For the second set of the thresholds, we define them to be ten times smaller than the differences obtained using a reduced time-step. We will refer to the first of the two cases as the tight case and the second as the loose case.

* Page 16, lines 4-5: this sentence does not make sense to me; please check it

| Variable | Accuracy score half-time step | Thresholds tight case | Thresholds loose case |
|---|---|---|---|
| Temperature (3D field) | $1.49 \times 10^{-3}$ | $1.49 \times 10^{-6}$ | $1.49 \times 10^{-4}$ |
| Salinity (3D field) | $6.41 \times 10^{-3}$ | $6.41 \times 10^{-6}$ | $6.41 \times 10^{-4}$ |
| Heat Content (2D field) | $2.74 \times 10^{-3}$ | $2.74 \times 10^{-6}$ | $2.74 \times 10^{-4}$ |
| Salt Content (2D field) | $5.47 \times 10^{-6}$ | $5.47 \times 10^{-9}$ | $5.47 \times 10^{-7}$ |

**Table 1.** Accuracy score of the simulation performed using a time-step shorter than the reference and the different thresholds used for the different variables in the tight and loose cases.

Having the reference simulations and the thresholds defined, we can already evaluate if the results of a simulation are accurate or not.



| Processor | 2 sockets Intel Xeon Platinum 8160 CPU with 24 cores each @ 2.10GHz for a total of 48 cores per node |
|---|---|
| Memory | L1d 32K; L1i cache 32K; L2 cache 1024K; L3 cache 33792K 96 GB of main memory 1.880 GB/core |
| Network | 100 Gbit/s Intel Omni-Path HFI Silicon 100 Series PCI-E adapter 10 Gbit Ethernet |
| Local Storage | 200 GB local SSD available as temporary storage during jobs |

**Table 2.** Marenostrum 4 node specifications

### 3.1.3 Executing the tests

To execute the tests, we implemented the simple rules described in the methodology section developing a simple python workflow manager capable of handling the dynamic workflow that will be creating the simulation scripts, launching them in a remote platform, checking the status and the adequacy of the results. We used python and the object-oriented paradigm to

create the methods that move the jobs from one status to another, making the code smooth and easy to follow. The key point of the tool is that given a variable set, it can generate the corresponding list of parameters used to select the precision used in the simulation, upload it to the remote machine, launch a simulation, monitor its status, evaluate the results to assess if these are accurate and also create subsets that will be also evaluated. To use the tool, we need to have configured a passwordless ssh acces to the remote machine, a script to execute a simulation with a given set and a script to compute the accuracy score of the

results produced by the simulation.

The perform the simulations, the Marenostrum 4 supercomputer has been used (see table 2). Each individual simulation using the emulator took about 8 minutes 30 seconds using a single node. While the proposed method has to start only with a single simulation, when the algorithm divides the sets there are more jobs that became available and might be executed in parallel, being possible to take advantage of having a large HPC system.

During the implementation of the emulator, the declarations of more than 3500 real variables were replaced with emulator variables. These variables can be scalars or arrays of up to 5 dimensions. They can be global variables or just temporary variables used only in a single line of code inside a subroutine. In a large code like NEMO there are usually parts of the code that are either used or not depending on the specific parameters employed. For this reason even when more than 3500 variables were identified, using our specific configuration only 942 are used. Given that considering the unused variables for the analysis

would be useless and dangerous (as we may be drawing wrong conclusions about which precision is needed for those unused variables) those are simply kept outside the analysis. The initial set used to start the analysis will be therefore the set containing all the variables that are actually used during our specific case.

### 3.1.4 Discussion

Starting with the evaluation of the initial set containing all the variables used in the simulation chosen for this illustration, the

results largely exceed both thresholds, showing that a global reduction of the precision used changes the results beyond what is acceptable. Following the algorithm, both cases keep dividing into smaller subsets and until the level 4 we do not find the first difference in the two cases. At this point, one of the subsets result presents a difference lower than the loose threshold





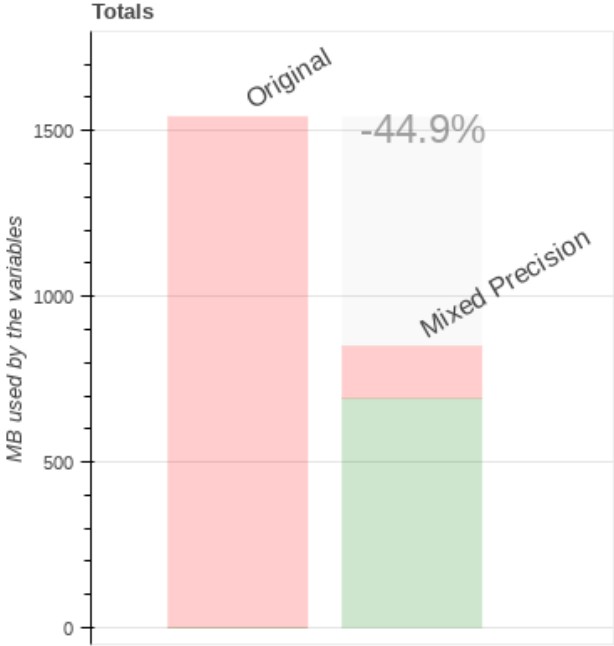

**Figure 1.** Estimation of the memory usage impact in an ORCA2 configuration using the set obtained with the loose accuracy test. The estimation has been performed having into account the dimensions of the variables and the size of the domain. In light red we have the memory that is occupied by variables in double precision (100% in the original case), in light green the proportion of the memory occupied by variables in single precision and in light gray the difference between the original case and the loose case, representing a potential 44.9% decrease in memory usage.

but still higher than the tight one. After this point, the paths diverge in several other places, and as could be foreseen, defining higher thresholds involves that more tests will be needed since there are more variables whose induced error is higher than the threshold. Comparing the two cases this seems to be true since the tight case implied 1442 simulations to arrive to the solution (consuming 204.3 node-hours) and the looser case just 321 (consuming 45.5 node-hours).

5    In the tight case, from the 942 variables that took part in the analysis, 652 variables (69.2%) were selected for single-precision, keeping the simulation error under the threshold. In the other hand, in the loose case the obtained solution contained 902 variables (95.8%), keeping only 40 variables in higher precision. As expected, defining looser thresholds allows to use lower-precision in a bigger proportion of the variables. Table 3 presents the results split by the dimension of the arrays.

Figure 1 represents the expected impact in the memory usage when using the set obtained with the loose accuracy test.

10 **3.2 ROMS**

The Regional Ocean Model System (ROMS) is a free-surface, hydrostatic, primitive equation ocean model that uses stretched, terrain-following coordinates in the vertical and orthogonal curvilinear coordinates in the horizontal. It contains a variety of fea-



| Dimension | Total variables | Tight case | | Loose case | |
|---|---|---|---|---|---|
| | | n | % | n | % |
| 0 | 374 | 304 | 81.3 | 366 | 97.9 |
| 1 | 32 | 23 | 71.9 | 32 | 100.0 |
| 2 | 322 | 216 | 67.1 | 307 | 95.3 |
| 3 | 200 | 105 | 52.5 | 190 | 95.0 |
| 4 | 13 | 3 | 23.1 | 6 | 46.2 |
| 5 | 1 | 1 | 100.0 | 1 | 100.0 |
| TOTAL | 942 | 652 | 69.2 | 902 | 95.8 |

**Table 3.** The table presents the total number of variables that were included in the analysis split by the dimension of the arrays (0 for scalars). Also, the number of variables that can safely use single-precision for the two cases discussed in section 3.1 (tight and loose) and the percentage of the total variables that them represent. Using tight constrains a 69.2 % of the variables can use single-precision, although its only a 52.5 % for the 3D variables that are responsible for the most memory usage. Using loose conditions, a 95.8% of the variables can use single-precision, being also a 95% of the 3D variables.

tures including high-order advection schemes; accurate pressure gradient algorithms; several subgrid-scale parameterizations; atmospheric, oceanic, and benthic boundary layers, biological modules, radiation boundary conditions, and data assimilation (see http://www.myroms.org).

The experiments performed with ROMS were done applying the primal form of incremental strong constraint 4D-Var
(4DVAR). The configuration used is the U.S. west coast at 1/3 degree horizontal, referred to as WC13, a standard model test case. It has  30 km horizontal resolution, and 30 levels in the vertical (see http://www.myroms.org). This configuration was selected because a 4DVAR ROMS has a large community of users, there is an easy-to-follow tutorial to set-up the configuration and involves linear models that make this an interesting choice to expand the results obtained in Section 3.1.

As is described in Moore et al. 2011, data assimilation is used in meteorology and oceanography to combine observations
and numerical models to obtain the best linear unbiased estimate of the circulation, and for other related applications, such as parameter estimation. The circulation estimates are usually defined as those which minimize, in a least-squares sense, the difference between the model state and the observations and a background (or prior) subject to a priory hypotheses about errors in the background, model, and observations.

To perform this exercise, we set-up the tutorial available online that performs a I4DVAR data assimilation cycle that spans the
period 3-6 January, 2004. The observations assimilated into the model are satellite sea surface temperature, satellite sea surface height in the form of a gridded product from Aviso, and hydrographic observations of temperature and salinity collected from Argo floats and during the GLOBEC/LTOP and CalCOFI cruises off the coast of Oregon and southern California, respectively.

With the I4DVAR version of ROMS, most of the computational time is spent inside the data assimilation "inner loops", within the adjoint and tangent linear sub-models (AD and TL respectively). To put some numbers, in a simulation with the
WC13 configuration, 35% of the time is spent in the TL model and 50% in the AD model while the time spent in the nonlinear




model is below 14%. It was felt that since the TL and AD models are used to compute an approximation gradient of the I4DVAR cost function, further approximations in the gradient resulting from lower-precision will probably not be so detrimental leading to our starting hypothesis that the AD and TL models are better targets than the nonlinear model. For this reason, our application of the methodology proposed in this manuscript is focused on trying to minimize the precision used in these regions of the

code. In this case, the target reduced precision is not limited to single-precision (23-bit significand) but also the half-precision is explored (10-bit significand).

### 3.2.1 Emulator Implementation

The strict style-guide of the ROMS code made that the same tool developed to implement the emulator in NEMO was quite straightforward to adapt for ROMS.

One aspect that made this exercise different from that of NEMO in terms of implementing the emulator was that the interest was focused on specific regions of the code, the parts related to the tangent linear model and the adjoint model. The complexity of leaving parts of the code using usual real variables instead of emulator variables is mostly in the interfaces between routines.

### 3.2.2 Designing the tests

When running a data assimilation experiment, the objective is to obtain a coherent state of the ocean that minimizes the

difference between the model and the observations. Through different forward-backward iterations using the TL and AD models, the model should converge to a state where the cost function (function that describes the difference between the model state and the observations) is minimum. We can consider that the model is accurate enough if the model shows the same convergence behaviour, i.e. the cost function ends up with the same value. Through the different iterations, the solution should converge to a minimum. To set a threshold for the accuracy of the simulations, we can look at the difference between the

last two inner-loop iterations. In the reference simulation this value is of $1.77 \times 10^{-1}$ . Defining a threshold 10 times smaller $(1.77 \times 10^{-2})$ ensures that the impact of reducing the precision will be smaller than changing the number of inner iterations performed.

### 3.2.3 Executing the tests

For ROMS, we could use the same workflow manager developed for NEMO, simply using different templates for launching

simulations and evaluating the results.

### 3.2.4 Discussion

Like in the NEMO case, to begin with the analysis we need to first perform reference executions using 64-bit precision for all the different initial conditions to have something to compare with. Again, the necessary tools prepared for NEMO were straightforward to adapt for ROMS, making the process fast and efficient.





| Dimension | Total variables | 23-bits | | 10-bits | |
|---|---|---|---|---|---|
| | | n | % | n | % |
| 0 | 326 | 326 | 100.0 | 277 | 85.0 |
| 1 | 82 | 82 | 100.0 | 71 | 86.6 |
| 2 | 450 | 450 | 100.0 | 385 | 85.6 |
| 3 | 173 | 173 | 100.0 | 135 | 78.0 |
| 4 | 78 | 78 | 100.0 | 44 | 56.4 |
| 5 | 31 | 31 | 100.0 | 9 | 29.0 |
| 6 | 4 | 4 | 100.0 | 2 | 50.0 |
| 40 | 2 | 2 | 100.0 | 2 | 100.0 |
| TOTAL | 1146 | 1146 | 100.0 | 925 | 80.7 |

**Table 4.** The table presents the total number of variables that were included in the ROMS analysis split by the dimension of the arrays (0 for scalars). Also, the number of variables that can safely use reduced-precision for the two cases discussed in section 3.2 (single and half precision ) and the percentage of the total variables that them represent. The single-precision case was trivial since all the variables considered can use it instead of double precision keeping the accuracy. For the half-precision case, an 80.7% of the variables could use half-precision, although the proportion of variables that can use reduced precision is minor in high-dimensional arrays than in the low-dimensional ones.

In ROMS, the analysis was focused only in the Tangent Linear and Adjoint models that are used to perform a I4DVAR data assimilation. The TL and AD parts plus the mod_ocean.f90 of the model contain 1556 real variable declarations, of which 1144 are used in the specific case used for this analysis. Starting with this initial set and trying to use a 23-bit significand for the variables with reduced precision, the results showed that actually all the variables contained in the parts studied (TJ and AJ models) can use reduced precision and still obtain results that are accurate according the test defined above.

This suggests that this model and in particular this specific configuration are suitable for a more drastic reduction of the numerical precision. To test that, we proceeded with a new analysis using a 10-bit significand for the variables with reduced precision. The first simulation of the analysis with all the variables of the set using reduced precision crashed due to an internal inconsistency, showing that it is not possible to reduce the precision overall. On further investigation, we found that by splitting the initial set we find that only one of the subsets makes the model crash. In this case we could see that this behaviour occurs in a single branch of the subsets until the only variable that makes the model crash when using reduced precision is identified. At this point of the analysis, we could see that it was in fact a single variable that was preventing the simulation to finish using a 10-bit significand in the TL and AD models. Nevertheless, many other variables introduce differences that are above the defined threshold. When the whole algorithm finishes, the resulting set contains a 80.7% of the variables, showing that it is possible to achieve accurate results reducing the number of bits from 52 to 10 for a big proportion of the code. In Table 4 the results are presented splitting by dimensions.



### 3.3 Common discussion

Looking at the two experiences shown in section 3 we can gather several conclusions. The most important conclusion is that the method can provide a set of variables that can use reduced numerical precision keeping the accuracy of the results, as it was observed in the four cases explored (two different accuracy tests for NEMO and two target precisions for ROMS). It is

also important to notice that the automation at several levels makes this method useful and viable. Without automation the full implementation of the emulator would require months of work and without the automation of the analysis workflow, the management of all the simulations needed would be unaffordable too. The results in the four cases showed that the room for reducing the numerical precision is considerable even in the most constrained cases (NEMO tight). While in the NEMO cases the analysis covers the full model, the ROMS case was constrained to only a part of the model, showing the versatility of the

method that can be used also to analyze a very specific part of the model with special insight and reduced cost. It has been shown also that it is possible to achieve simulations that are arbitrarily close to the reference simulation, where requiring closer results will have the cost of leaving more parts of the code in high precision.

### 4 Conclusions

Previous works suggested that the generalized use of double precision for Earth Science modelling is a case of over-engineering

and that for this reason adapting the computational models to use the numerical precision that is really required would pay back in terms of performance improvement. However, an improper reduction can led to accuracy losses that may make the results unreliable. In this manuscript we presented a method that was designed to solve this problem by finding which variables can use a lower level of precision without degrading the accuracy of the results.

The method was thought and designed to be applied to computational ocean models coded in Fortran. It relies on the Reduced

Precision Emulator tool that was created to help understand the effect of reduced precision within numerical simulations, it allows us to simulate the results that would be produced in case of performing the arithmetic operations in reduced precision, using for each floating point variable a custom number of significant bits. The proposed analysis algorithm finds which variables need to be secured in double precision and which ones can use reduced precision without impacting the results accuracy. The algorithm is based on a binary search algorithm and not only identifies the variables that must be secured in double-precision

but builds a reliable set of variables proven to provide accurate results.

The method has been put to the test with two widely used state-of-the-art ocean models, NEMO and ROMS. The experiences with the two models pursued different objectives. With NEMO, the analysis covered all the routines and variables used with the ocean-only simulation, the target precision was 32-bits and we explored how the selection of a specific accuracy tests changes the variables that can safely use reduced precision. With ROMS instead, the analysis covered only the variables belonging to

the adjoin and tangent linear models, we used a single accuracy test and we tried out how the method can be used to examine the viability of using numerical precisions below 32-bits.

The results presented in this work allows us to drive to some conclusions. Regarding the models, it is shown that both can use reduced precision for large portions of the code proving the feasibility of mixed precision approaches in these two ocean





models. Regarding the the method described in this manuscript, it proved to be capable of finding the large parts of the code that can use reduced-precision arithmetic keeping the accuracy of the simulations. Its versatility is also proven being able to provide useful results for four different experiences using two different models. It can be used for a full model analysis or instead for analyzing a specific region of the code, reducing the computational resources needed. It was proven also that the

method can provide a configuration that can be arbitrarily close to the double-precision reference, depending on the amount of variables that can use reduced-precision on how strict the conditions imposed are.

The logical step that will follow the work presented here is to obtain feedback from the community and build consensus about the conditions to consider that the results are sufficiently accurate. Users might want to follow the method presented here to build a version of the model that can benefit from the reduction of the precision and can be used by all the community

without concerns.

*Code and data availability.* Releases of the Reduced Precision Emulator library are available at https://github.com/aopp-pred/rpe (last accessed in January 2019). The tool developed to automatically implement RPE can be found at https://earth.bsc.es/gitlab/otinto/AutoRPE. The workflow manager used to manage the analysis simulations can be found in https://earth.bsc.es/gitlab/otinto/AutoRPE. The source code of NEMO model and the input data used for the experiments referred in this work can be found in www.nemo-ocean.eu (last accessed in

January 2019). The source code of the ROMS model can be found in www.myroms.org, can be freely accessed under registration. The details and input data used for the experiments involving ROMS can be found in www.myroms.org/wiki/I4DVAR_Tutorial (last accessed in January 2019).

*Author contributions.* Oriol Tintó Prims led the design of the method, the experiments and the writing of the manuscript, Mario C. Acosta supervised all the process and had a main role in the writing of the manuscript, Andrew M. Moore participated in the design of the experiments

involving ROMS and the writing of the manuscript, Miguel Castrillo was involved in the discussion and design of the method and the experiments and participated in the writing of the manuscript, Kim Serradell, Anna Cortés and Francisco J. Doblas-Reyes participated in the discussions and the revision of the manuscript.

*Acknowledgements.* The research leading to these results has received funding from the EU ESiWACE H2020 Framework Programme under grant agreement no. 675191, from the Severo Ochoa (SEV-2011-00067) programme of the Spanish Government and from the Ministerio de

Economia y Competitividad under contract TIN2014-53234-C2-1-R.





## Appendix A

---

**Algorithm 1** The analysis algorithm describes the actions that have to be taken in a given set that has a given status. Using that framework, the list of possible statuses is: **Pending**: the set has is ready to be evaluated, **Running**: the set is being evaluated, **Asserting**: the set had a successful evaluation and it is ready to be reevaluated using a different configuration, **Suspended**: the set had an unsuccessful evaluation and now it is waiting for the results of its child subsets, **Failed**: the set had a unsuccessful evaluation and can not be further divided.

---

```
if STATUS is PENDING then
    submit job
    set STATUS to RUNNING
else if STATUS is RUNNING then
    Check job status in remote machine
    if SIMULATION was SUCCESSFUL then
        Assert results and set STATUS to ASSERTING
    else
        if the set can be divided then
            Create and submit sub-sets and set STATUS to SUSPENDED
        else
            Set STATUS to FAILED
        end if
    end if
else if STATUS is ASSERTING then
    Check assertion job status in remote machine
    if job status is COMPLETED then
        Check results:
        if results are successful then
            Set STATUS to SUCCESS
        else
            if  set can be divided then
                Create and submit sub-sets and set STATUS to SUSPENDED
            else
                Set STATUS to FAILED
            end if
        end if
    end if
else if STATUS is SUSPENDED then
    Check subsets STATUS
    if subsets are FAILED or SUCCESS] then
        if Both subsets FAILED then
            Set STATUS to FAILED
        else
            Integrate subset information and set STATUS to PENDING
        end if
    end if
end if
```

---



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
