# Peer review of "How to use mixed precision in Ocean Models"

_Geoscientific Model Development, 2019_

## Referee Comment (RC1) · Peter Düben (Referee) · 10 Mar 2019

This paper is summarising a significant amount of work and provides several important contributions: it automates the introduction of a reduce precision emulator into a complex ocean model; it presents a method to optimise numerical precision automatically within such a model; it provides a rough estimation of memory savings if precision is reduced. The results are very relevant for GMD and the paper should be published. However, a revision following the comments below may improve the paper.

Major comments:

- There is one weakness of the presentation at the moment: Results for model simulations that are using the reduced precision configuration are not presented. If these

are not shown, the reader will assume that the results are not so great but I think that this is actually not the case. Can you add some figures showing results when using the reduced precision configurations? E.g. mean fields for long term simulations and differences for short term simulations in comparison to differences that are caused by a change of the timestep?

- The English should be revised and improved throughout the paper (see some detailed comments below but there are more problems in the text).

Minor comments:

- One of the main problems for a precision reduction in ocean models is that conservation laws may get violated (mass and tracers). Can you comment on that? Or quantify mass loss/gain when running the reduced precision configurations?

- The constraints that come in via the exponent are not very well discussed.

- You may want to cite this paper: Düben, P. D., A. Subramanian, A. Dawson, and T. N. Palmer (2017), A study of reduced numerical precision to make superparameterization more competitive using a hardware emulator in the OpenIFS model, J. Adv. Model. Earth Syst., 9, 566–584, doi:10.1002/2016MS000862. It performs a precision analysis per parameter for the CRM used in superparametrisation similar to the one performed in this paper. It is also arguing that the parameter uncertainty that is found via an automated precision analysis could be used to develop stochastic parametrisation schemes. This may also add to an interesting discussion in this paper.?

- Page 4, second paragraph, "The objective..." This paragraph is difficult to understand and you may want to revise this. It may be easier to explain this with an example.

- Page 4 l31: "any information" is too strong.

- Page 5 l5: "uncertainty in the model inputs" but also model error.

- Page 8: ll 12-22 are very difficult to understand.

- Page 10: I do not understand IQR and how it is used.

- Figure 1: Why do you not present the results for the tight evaluation? The light cray is not visible in my printout.

- For NEMO: Maybe I have missed this information: Do you state how many of the variables are actually used and how many are not used in standard simulations? You should also state the accuracy score when you run simulations with the combined precision reductions for the tight and loose precision configuration.

- Page 15: "AD and TL should be better targets" 4DVar experts tend to disagree with this statement since the forward TL and the backward AD need to fit to each other to guarantee convergence of the assimilation. Reduced precision can destroy this. You may want to discuss this less optimistically.

- Page 16 l12: Can you state what the "single variable" is?

- Page 16 l14: "80.7%" of the variables: Is this in amount of variable declarations or the amount of bits stored in memory? (This is probably stated somewhere but I could not find it easily).

Language:

Abstract: ll 13-15: "have the potential" should be revised. This part of the text is difficult to understand.

- Page 4 l33: "were accurate resulted in an inaccurate set when combined" should be rephrased.

- Page 4 l3: "for sure consider that" should be rephrased.

- Page 7 l27: "potential error" should be rephrased.

- Page 8 l3: "real arithmetic" -> "real number arithmetic"

- Page 8 l26: "a target reduced precision single-precision" should be rephrased.

- Page 8 l27: "analysis" -> "analyses"

- Page 9 l4: "the biogeochemistry" -> "biogeochemistry"

- Page 9 l9: "a 20%" -> "20%"

- Page 9 l10: "needs to be something that does not" -> "cannot"

- Page 9 l29: "netCDF, the used by" should be rephrased.

- Page 12 l3: "that will be creating the simulaton scripts" should be rephrased.

- Page 12 l26L "and until the level 4 we do" is unclear.

- Figure 1: "having in the account" should be rephrased.

- Page 13 l6: "In the other hand" -> "On the other hand"

- Table 3: "that them represent" should be rephrased.

- Page 15: "made that the same tool" should be rephrased.

- Table 4: "that them represent" -> "that they represent". "an 80.7%" -> "80.7%"

- Page 16 l3: "mod_ocean.f90" is a bit out of context.

- Page 17 l2: "looking at the two experiences" should be rephrased.

- Page 17 l16: "can led to" -> "can lead to"

- Page 18 l1: "the the"

- Page 19: "the set has is"

---

## Short Comment (SC1) · 14 Mar 2019

Dear authors,

In my role as Executive editor of GMD, I would like to bring to your attention our Editorial version 1.1:

http://www.geosci-model-dev.net/8/3487/2015/gmd-8-3487-2015.html

This highlights some requirements of papers published in GMD, which is also available on the GMD website in the 'Manuscript Types' section:

http://www.geoscientific-model-development.net/submission/manuscript_types.html

In particular, please note that for your paper, the following requirements have not been

met in the Discussions paper:

- "The main paper must give the model name and version number (or other unique identifier) in the title."

- "If the model development relates to a single model then the model name and the version number must be included in the title of the paper. If the main intention of an article is to make a general (i.e. model independent) statement about the usefulness of a new development, but the usefulness is shown with the help of one specific model, the model name and version number must be stated in the title. The title could have a form such as, "Title outlining amazing generic advance: a case study with Model XXX (version Y)"."

- "All papers must include a section, at the end of the paper, entitled 'Code availability'. Here, either instructions for obtaining the code, or the reasons why the code is not available should be clearly stated. It is preferred for the code to be uploaded as a supplement or to be made available at a data repository with an associated DOI (digital object identifier) for the exact model version described in the paper. Alternatively, for established models, there may be an existing means of accessing the code through a particular system. In this case, there must exist a means of permanently accessing the precise model version described in the paper. In some cases, authors may prefer to put models on their own website, or to act as a point of contact for obtaining the code. Given the impermanence of websites and email addresses, this is not encouraged, and authors should consider improving the availability with a more permanent arrangement. After the paper is accepted the model archive should be updated to include a link to the GMD paper."

As the tool you have developed and you are publishing with the article is RPE, this should be named (including version number) in the title of the manuscript. e.g. "Analysing how to use mixed precision in Ocean Models using RPE (V x.y)"

Note, that the exact version of the code/tools described in an article should be **permanently** accessible. Thus please consider to make the exact version, your article refers to, available via a permanent archive providing a DOI (e.g. Zenodo).

Yours,

Astrid Kerkweg

---

## Referee Comment (RC2) · Matthew Chantry (Referee) · 15 Mar 2019

The authors examined the use of reduced numerical precision to ocean modelling. In particular they examine the NEMO and ROMS ocean models and use a divide-and-conquer algorithm to establish which variables can be calculated with reduced numerical precision. Overall this was excellent and interesting science and a useful tool for investigating precision reduction has been developed. I have made a few points below which I would like the authors address. However, once finalised this paper will be a valuable contribution to GMD.

Overall assessing quality of simulations using RMS (or similar as used here) is a good approach. However it is useful to pair this with some representative map plots (or other

data representation) which can give the readers a sense for the locations and scales of the errors. I would recommend the authors introduce these figures, particularly for the NEMO section.

The authors reference multiple simulations for the NEMO runs. Could they clarify when they use multiple initial conditions and how sensitive the tests are to the initial conditions.

The paper is hindered by its length, there are many points where too many words are used to communicate the necessary information. Cutting space can be a challenging task but the final product will be a paper where readers can easily digest all the information that the authors wish to communicate. Ideally the authors would run through the text and ask themselves whether there is a more concise way of communicating each point or whether the information needs to be communicated. I have highlighted just a few examples in my minor points below.

Minor points.

* P7:S2.2 Designing accuracy tests. What is the purpose of this section? This question is more thoroughly discussed in the model-specific sections. I would advise removing this section.

* P8:L3-7 Unsure of the purpose of this example. Suggest removing.

* P8:L8-14 This paragraph was unclear, could the authors please reformulate.

* P9:L15 Do you have a citation for this?

* P10 How are the quartiles defined? Over space?

* How does the ratio of RMSD and IQR vary as a function of time generally? Are failed simulations more likely to exceed the threshold for early/late times?

* P11:L10 "smaller time-step" -> "halved time-step"

\* Example of text reduction. P11 the variable values are duplicated in text and the table below. I would recommend removing from the text and referring to the table.

\* P11 Table: Column headings for thresholds could include their definition. e.g. Tight threshold = 10ˆ(-3)

\* Accuracy Score.

\* P12:L15 How are the variables that do not impact the results identified?

\* P13:F1 The colour scheme used is not printer-friendly. The figure caption could be reduced in length if a key/legend was provided for the figure. Why is a bar for the tight constraint not plotted?

\* P13:L1 "After this point . . ." This sentence is too long and I couldn't understand what it was trying to communicate.

\* P16:L8-13 Another example ripe for text reduction. Simply stating that a single variable was responsible for breaking the procedure would suffice. Which variable was responsible?

\* P16 Table. Could the authors reproduce Fig1 but for results with both single and half-precision for ROMS.

\* P21:L1 Typo "wheather" -> "weather"
* * *

---

## Short Comment (SC2) · 20 Mar 2019

There has been a lot of work on multiprecision scientific computing since the Baboulin et al. paper in 2009. A recent paper that could be cited is

```
@inproceedings{htdh18,
  author = "Azzam Haidar and Stanimire Tomov and Jack Dongarra and
            Nicholas J. Higham",
  title = "Harnessing {GPU} Tensor Cores for Fast {FP16} Arithmetic to Speed up
           Mixed-Precision Iterative Refinement Solvers",
  booktitle = "Proceedings of the International Conference for High
               Performance Computing, Networking, Storage, and Analysis",
```

[Figure]

```
  year = 2018,
  series = "SC '18 (Dallas, TX)",
  pages = "47:1--47:11",
  address = "Piscataway, NJ, USA",
  publisher = "IEEE Press",
url = "http://dl.acm.org/citation.cfm?id=3291656.3291719"
}
```

---

## Author Comment (AC1) · 10 May 2019

We thank the consideration that the referee had for our work and the valuable comments that helped improve the quality of our manuscript. Answers for the questions follow this lines, and the specific suggestions to improve the text have been adopted.

**Referee:** There is one weakness of the presentation at the moment: Results for model simulations that are using the reduced precision configuration are not presented. If these are not shown, the reader will assume that the results are not so great but I think that this is actually not the case. Can you add some figures showing results when using the reduced precision configurations? E.g. mean fields for long term simulations and differences for short term simulations in comparison to differences

that are caused by a change of the timestep?
**Answer:**The reason to leave the results without a figure was that the maps are virtually identical and the differences are not perceptible. Figure 1 and Figure 2 attached to this document show the sea surface temperature mean during the first month of simulation for the reference case and the loose case. It can be seen that there are no perceptible differences. A paragraph has been added to the discussion section 3.1.4 trying to empathize this.+

**Referee:** The English should be revised and improved throughout the paper (see some detailed comments below but there are more problems in the text).
**Answer:** There has been an effort to improve the quality of the text, all the mistakes pointed out by the referee have been solved and a general revision of the text has been performed.

**Referee:** One of the main problems for a precision reduction in ocean models is that conservation laws may get violated (mass and tracers). Can you comment on that? Or quantify mass loss/gain when running the reduced precision configurations?
**Answer:** The conservation of mass and tracers in our experiments has not been studied. It is in fact a good idea to consider conservation laws to be used as accuracy tests as well. Defining an accuracy test to ensure that conservation laws are fulfilled will prevent that any variable that might negatively affect conservation is set to reduced precision.

**Referee:** The constraints that come in via the exponent are not very well discussed.
**Answer:** I have added clarifications regarding this point in two different sections of the code: when we first talk about the reduced precision emulator in the introduction and when we talk about the possibility of using half-precision for ROMS simulations in section 3.2 .

**Referee:** You may want to cite this paper: Düben et al. 2017: "A study of reduced numerical precision to make superparameterization more competitive using a hardware emulator in the OpenIFS model".It performs a precision analysis per parameter for the CRM used in superparametrisation similar to the one performed in this paper. It is also arguing that the parameter uncertainty that is found via an automated precision analysis could be used to develop stochastic parametrisation schemes. This may also add to an interesting discussion in this paper.?

**Answer:** A cite to this manuscript has been included in the introduction.

**Referee:** I do not understand IQR and how it is used.

**Answer:** Interquartile Range is a measure of statistical dispersion of the values of a sample. In the manuscript, it is used in the NEMO cases to normalize the RMSD minimizing the effects of outlier values. At first, it was considered because it would make it easier to combine errors from variables of different magnitudes, although in the case presented in this manuscript it might be unnecessary.

**Referee:** Why do you not present the results for the tight evaluation? The light cray is not visible in my printout.

**Answer:** The tight case was included to demonstrate that the method can work imposing different constrains. Showing the impact estimates for the tight case can mislead the reader to think that the potential reduction of memory usage is lower that what it really is. This same information has been included in the manuscript. The Figure 1 has been modified to make the two colors distinguishable in black and white.

**Referee:** For NEMO: Maybe I have missed this information: Do you state how many of the variables are actually used and how many are not used in standard simulations? You should also state the accuracy score when you run simulations with the combined precision reductions for the tight and loose precision configuration.

**Answer:** From more than 3500 variables declared through the code, our specific configuration only uses 942. These are identified during a simulation and a file that

shows which variables were actually used is generated. This information has been included in the manuscript (section 3.1.3). Our experience using the method with NEMO tells us that with a large number of variables, the final accuracy score ends up being just below the limit established. This has been empathized in the manuscript.

**Referee:** "AD and TL should be better targets" 4DVar experts tend to disagree with this statement since the forward TL and the backward AD need to fit to each other to guarantee convergence of the assimilation. Reduced precision can destroy this. You may want to discuss this less optimistically.

**Answer:** TL and AD models that are typically used in 4D-Var systems involve many approximations (e.g. different resolutions to the NL model, reduced physics, more stable numerics, etc) - nonetheless they provide useful gradient information. We see reduced precision as simply another level of approximation. As long as the TL and AD models are consistent (i.e. the AD will be the transpose of the TL to within the chosen precision), the numerical precision can be reduced. The referee was right pointing out that our previous explanation could seem too optimistic and it was clarified in the manuscript.

[Figure]

**Fig. 1.** Monthly mean sea-surface temperature in the first month of the reference simulation.

[Figure]

**Fig. 2.** Monthly mean sea-surface temperature in the first month of the loose case simulation.

---

## Author Comment (AC2) · 10 May 2019

We are thankful for the referee's positive feedback and the valuable comments that helped to improve the quality of the manuscript. Specific answers for the major comments follow this lines. The minor comments have been addressed and the corrections are included in the manuscript.

**Referee:** Overall assessing quality of simulations using RMS (or similar as used here) is a good approach. However it is useful to pair this with some representative map plots (or other data representation) which can give the readers a sense for the locations and scales of the errors. I would recommend the authors introduce these figures, particularly for the NEMO section.

[Figure]

**Answer:** We have added a paragraph in section 3.1.3 explaining how the magnitude of the differences is below the thresholds defined and therefore inappreciable in a regular plot.

**Referee:** The authors reference multiple simulations for the NEMO runs. Could they clarify when they use multiple initial conditions and how sensitive the tests are to the initial conditions.

**Answer:** As it is explained in section 2.3. , we were using different initial conditions to double-check the results. During the analysis, whenever an experiment was positively evaluated with the first set of initial conditions, it was re-evaluated with a second set of initial conditions to increase the confidence on the results. It has been clarified in the text.

**Referee:** The paper is hindered by its length, there are many points where too many words are used to communicate the necessary information. Cutting space can be a challenging task but the final product will be a paper where readers can easily digest all the information that the authors wish to communicate. Ideally the authors would run through the text and ask themselves whether there is a more concise way of communicating each point or whether the information needs to be communicated. I have highlighted just a few examples in my minor points below.

**Answer:** The authors have done an effort to reduce the length of the document. This included rephrasing several parts all along the document and the reviewed manuscript is three pages shorter. We believe that the quality of the document has improved and many parts have been simplified.

**Referee:** P7:S2.2 Designing accuracy tests. What is the purpose of this section? This question is more thoroughly discussed in the model-specific sections. I would advise removing this section.

**Answer:** Being able to determine if a result is accurate or not is a basic requirement for the analysis. The purpose of this section is to show a simple example that tries

to illustrate that not only the actual results but also how we evaluate them will impact the method outputs. Wee consider that it is important to keep it in the manuscript but however we shortened it.

**Referee:** P8:L3-7 Unsure of the purpose of this example. Suggest removing.
**Answer:** The example was provided because in a previous review it was considered useful to show a simple case in which a result of a arithmetic operation was wrong only when both variables involved were using reduced precision. We consider it illustrative for audience without an extensive experience in the field.

**Referee:** P9:L15 Do you have a citation for this?
**Answer:** This affirmation is based in our own experiments but the results have not been published.

**Referee:** How are the quartiles defined? Over space?
**Answer:** For each output variable and for each time-step the quartiles are computed using the data of the full spatial domain.

**Referee:** * How does the ratio of RMSD and IQR vary as a function of time generally? Are failed simulations more likely to exceed the threshold for early/late times?
**Answer:** Since the simulations with reduced precision start with the same initial conditions than the reference, the RMSD at the beginning is very small and grows with time. The specific trajectories depend on the output variable analyzed, the variables which precision is being reduced and other casuistic. There is interesting information that can be learned from the results of a precision analysis, for instance the specific impact of each variable that is determined to require double-precision. These kind of analyses, however, go beyond the scope of this manuscript and might be part of future work.

---

## Author Comment (AC3) · 10 May 2019

We thank the editor to help us to adjust to the journal requirements. From the points exposed, we would like to do the following remarks.

- The contributions of the manuscript go beyond an individual model. If it is required to specify in the title which models have been explored in the document, the final title would be: *"How to use mixed precision in Ocean Models: exploring potential reduction of numerical precision in NEMO 4.0 and ROMS 3.6"*

- There has been a misunderstanding. Although we used the RPE tool. It is not the main contribution of the paper. There are few tools that have been devel-

oped and already appear in the code availability section. These tools are publicly accessible, as well as the other sources cited.

---

## Author Comment (AC4) · 10 May 2019

We thank the time invested reading the manuscript and the contribution to improve its quality. We did not know beforehand the work mentioned but after reading it we consider that it is appropriate to add the reference since there is a good amount of work that goes along the arguments exposed in the text.

---

## Author Response (AR2)

Dear Dr. Phipps,
many thanks for your remarks.

The manuscript now includes a new figure and also a supplementary material to sustend one of the reference requirements, if the editor thinks that a different way to present this information can be better we will be happy to modify it.
The following pages provide a point-by-point reply to the comments and a marked up version of the manuscript.

Sincerly,
The authors

However, I would like to expand upon one of the comments by the reviewers first. Specifically, one of the reviewers raises the issue of conservation of mass and tracers. This issue is of fundamental importance, particularly when using ocean models to conduct very long simulations (such as for palaeoclimates). In your response, you state that this issue has not been studied.

I consider that this issue needs to be acknowledged in some form in the manuscript. At an absolute bare minimum, it should be discussed in the text. However, I would like you to present results if at all possible. Presumably it is possible to assess any conservation errors in the simulations that you ran? Even if you cannot calculate conservation errors directly, it is presumably possible to compare the evolution of global-mean salinity (etc.) between the simulations with full precision and reduced precision?

I suspect, because of your accuracy tests, that the reduced precision does not introduce significant conservation errors. However, it would strengthen your manuscript to confirm this by quantifying the magnitude of any errors in conserving mass or tracers.

As it was pointed out, a reviewer raised the concern about the issue of trace and mass conservation. Our response was:

"The conservation of mass and tracers in our experiments has not been studied. It is in fact a good idea to consider conservation laws to be used as accuracy tests as well. Defining an accuracy test to ensure that conservation laws are fulfilled will prevent that any variable that might negatively affect conservation is set to reduced precision."

The part that was included in the manuscript did not talk directly about conservation and probably was not clarifying enough:

"The method ensures that the differences between the reduced-precision simulations performed with the final variable sets and the reference are below the determined thresholds"

What we wanted to remark is that the proposed method guarantees that the conditions used to discriminate between accurate and inaccurate results are fulfilled. However, the conditions (or accuracy test) used to exemplify how the method can be used with NEMO might not be the best to ensure that the resulting set will be useful for all the applications using NEMO.

However, as it seems that the specific results regarding conservation of tracers for the specific example can be of interest, a figure has been added to the manuscript comparing the daily global heat content and the daily global salt content for the first month of simulation, showing the values for the reference in double precision, the case with single precision and the case using the loose configuration found with the method described in the manuscript. Together with the plot, the following paragraph will be added:

"Although the accuracy test used for this example was not designed to ensure the conservation of global quantities, Figure X shows that the global heat and salt content of the simulations performed using the loose configuration resembles those of the reference

simulation performed in double-precision, which was not the case for a simulation fully performed using single-precision."

We value the insight of the editor and the reviewers because we realize that adding this information adds a lot of value to the documen. However, our intention is to leave more complete analysis and verification for the future work that will lead to a mixed-precision implementation of the NEMO.

**Original text:**
In the case of sea surface temperature measurements from Earth orbiting satellites, this precision to a few tenths of a degree.

**Correction:**
In the case of sea surface temperature measurements from Earth orbiting satellites, this precision is of a few tenths of a degree.

**Original text:**
 While it is true that this approach can give an insight about the precision needed by a particular variable, it has some drawbacks. This approach can possibly most of the variables that require high precision, however it may not reveal many issues regarding more complex interactions between more than one variable.

**Correction:**
While it is true that this approach can give an insight about the precision needed by a particular variable, it may not reveal issues regarding more complex interactions between more than one variable.

**Original text:**
This configuration was selected because a 4DVAR ROMS has a large community of users, there is an easy-to-follow tutorial to setup the configuration and involves linear models that make this an interesting choice to expand the results obtained in Section 3.1.

**Correction:**
This configuration was selected because a̶ 4DVAR ROMS has a large community of users, there is an easy-to-follow tutorial to setup the configuration and involves linear models that make this an interesting choice to expand the results obtained in Section 3.1.

**Original text:**
With NEMO, we performed two analyses using different

accuracy tests having as target to run using single-precision (23-bit significand).

**Correction:**
With NEMO, we performed two analyses using different
accuracy tests and both cases have as a target to find which variables can use
single-precision (23-bit significand).

**Original text:**
Previous experiences in reducing the working precision of
NEMO from 64-bit to 32-bit demonstrated a significant change in the results, indicating that
blindly reducing the precision in the entire code was not an option.

**Comment:**
The information supporting this affirmation is has not been published before and there's no
publication with this information. A supplementary material has been provided to support the
statement and a citation to this material has been included in the manuscript.

**Original text:**

[revised manuscript text omitted]

---

## Author Response (AR3)

Dear Dr. Phipps,

we are very thankful to have our manuscript accepted.

We are happy to improve the document following your suggestions:

The Figure 1 has been corrected, and the figure that was included in the supplementary material has now been included in the appendix B.

Additionally, the Figure in the appendix B has been modified to make it more easy to understand by inverting the color map to have the warm biases in red and the cold biases in blue.

Sincerely,

the authors